# Barriers to Effective Communication between Patients, Relatives, and Health Care Professionals in the Era of COVID-19 Pandemic at Public Hospitals in Limpopo Province

**Ndidzulafhi Selina Raliphaswa \***, **Dorah Ursula Ramathuba**, **Takalani Rhodah Luhalima**, **Seani Adrinah Mulondo**, **Thivhulawi Malwela**, **Azwidihwi Rose Tshililo**, **Julia Langanani Mafumo**, **Maria Sonto Maputle**, **Mary Maluleke**, **Takalani Ellen Mbedzi**, **Hilda Nwamuhohova Shilubane**, **Nthomeni Dorah Ndou**, **Khathutshelo Grace Netshisaulu** and **Mutshinyalo Lizzy Netshikweta**

Department of Advanced Nursing Science, University of Venda, Private Bag X 5050, Thohoyandou 0950, South Africa
\* Correspondence: ndidzulafhi.raliphaswa@univen.ac.za

**Abstract:** Background: Effective communication between health care professionals, admitted patients, and their families is crucial to improving the health care outcomes and patient satisfaction. Hence, barriers to effective communication cause a lot of confusion, frustration, and misunderstanding. The study aimed to identify the perceived barriers to effective communication between patients, their families, and health care professionals during the COVID-19 pandemic in public hospitals in Limpopo Province. The study was conducted in rural areas of Vhembe District, Limpopo Province. Methodology: A qualitative exploratory descriptive method was adopted to explore and describe the barriers of effective communication among patients, relatives, and health care professionals during the COVID-19 pandemic in public hospitals in the Vhembe District. Eleven (11) participants were purposively selected. Unstructured in-depth individual interviews were used to collect data and analyzed using thematic analysis following Tesch's eight steps. Results: The study findings revealed that there was ineffective communication by health care professionals. This was discussed under three themes that emerged: poor communication of diagnosis and prognosis and treatment outcomes; lack of accurate knowledge regarding the virus morphology, variant, and treatment protocols; and the inconsistencies in the dissemination of COVID-19 protocols. Conclusions: Full communication benefits both patients, relatives, and health care professionals as knowledge and understanding are enhanced through communication. Therefore, health care professionals should provide the patients and their relatives with full information about the disease and the visitation protocols to be followed. The study contribution was to explore barriers to effective communication to the patients and relatives by the health care professionals.

**Keywords:** barrier; communication; COVID-19; effective; pandemic

## 1. Introduction

As the world continued to battle coronavirus disease (COVID-19), the effects of its protocol restrictions have negatively impacted communication with patients [1]. Communication is an open dialogue between two parties, and is the process of sending and receiving messages through verbal or non-verbal means including speech or oral *communication*. Through communication, one can evaluate or assess whether communication is open and frank, and facial expression says a lot in communication [2]. Sharma and Choudhury (2021) were also of the opinion that non-verbal behavior plays a significant role in the quality and satisfaction of the patient–health worker relationship, which in turn influences adherence to the treatment program, resulting in positive clinical outcomes. Wearing face masks affects communication as it conceals the health care professionals' face and affects the tone of voice; it also increases the emotional trauma when the discussion is about the patient

and not with the patient, and communication is in the language that is not primarily used by the patients [3]. The effects of social distance also negatively affected these relationships because everyone feared the spread of infection [4]. The health care professionals were also anxious and uncertain, and the examination of patients, touch, palpation, percussions, auscultation, and the consultation length were severely compromised because of the fear of contagion.

Communication can also be written, however, during the COVID-19 pandemic, it became very non-human, as it was supposed to open up communication. Consequently, health care professionals relied on written notes, which is non-human in a way. Furthermore, patients referred from treatment centers to hospitals felt alienated by health care professionals as they only engaged with the medical notes rather than the individual patients and relatives who were present to gather more data from them. These resulted in patients and relatives experiencing the emotional and psychological trauma of not being involved in the care [5]. Albahri, Abushibs, and Abushibs (2018) indicated that other communication barriers among health care professionals and patients were difficulties with rapport building, physicians not explaining the condition and management effectively to their patients, the language and the culture of the patient and physician, and the physical setup of the clinic, all of which are potential barriers that have been shown to affect communication between physicians and patients.

The Department of Health in South Africa developed the Patient's Rights Charter, which emphasizes that patients have rights. According to Chapter 2 of the Bill of Rights, patients have the right to information pertaining to their health and to express themselves and to have their freedom of expression respected, especially where their health care is concerned, and the right to converse in the language of one's choice where possible [6].

Miscommunication between patients, relatives, and health care professionals may result in negative patient outcomes and relatives being dissatisfied with the care provided [7]. Furthermore, with limited travel restrictions imposed, relatives could not access information about the patients' condition or prognosis due to the fact that when making a call to the hospital, ethically, health care professionals are not allowed to disclose such information via telephone. Therefore, this further contributed to the communication barriers. Health care professionals are obliged to protect the privacy and confidentiality of their patients and to only disclose health care, treatment, diagnostic, and other health information with the patient's informed and written consent. Unfortunately, at that moment, this was difficult to achieve as the relatives had the right to know about what was happening to their loved ones. In African culture, when a person is sick or ill, it is about collectivism rather than individualism as it is the extended family's responsibility to know and make collective decisions. Based on the COVID-19 protocol restrictions, the researchers wanted to understand the communication barriers experienced by patients and their relatives during the COVID-19 pandemic at public institutions in Limpopo Province.

## 2. Materials and Methods

A qualitative approach using explorative and descriptive design was used. The study was conducted in 2021 at Vhembe District in Limpopo (as a Departmental Project in Nursing Science). An explorative, descriptive qualitative approach provided a full narrative of what was experienced by the participants to further understand the topic being explored [8]. However, [9] briefly considered exploratory research, and suggested that it should be designed to illuminate how a phenomenon is manifested, and is especially useful in uncovering the full nature of a little-understood phenomenon.

### 2.1. The Study Setting

The study was conducted in four (4) selected public hospitals in Vhembe District, amongst the four selected public hospitals, one (1) was a referral from the district hospital of Limpopo Province.

### 2.2. Population

The study population was made up of patients who were admitted during the COVID-19 pandemic during March 2020 up to October 2021. The target population were relatives and all participants who had tested positive to COVID-19 and were admitted to selected hospitals in Vhembe District.

### 2.3. Sampling

Sampling is the process of selecting a sample from the population. The sample comprised four district public hospitals and one regional hospital. Purposive sampling was used to sample four district hospitals and one regional hospital. Public hospitals were chosen on the basis that they are the institutions most commonly used by the local communities as no fee/paying facilities, given that most patients in the region are medically uninsured. Participants were also selected deliberately as they were to provide information-rich cases for an in-depth study, and they had experienced or were known to possess special experience to provide the information the researchers needed to answer the research question. The researchers envisaged sampling 20 participants, which was five from each district. However, only eleven participants were sampled as the data saturated before reaching the estimated number. Only patients who were admitted and were confirmed to be COVID-19 positive were traced using their address and admission book at the selected hospitals. This was conducted in order to access the patient's name, contact numbers, physical address, and the next of kin or relative. Permission to access the hospital admission book was granted by the managers of the selected hospitals.

### 2.4. Ethical Considerations

Ethical considerations were insured by obtaining permission (SHS/20/PDC/19/0608) from the University of Venda's Ethics Committee as well as permission to conduct the study from Board members of the hospitals and the participants. Participants were coded as participant number for each participant. However, the real names were not included in the voice recorder to enhance anonymity and confidentiality. Participants gave verbal consent as the research was conducted via telephone and included in the voice recording. The participants were informed of their right to withdraw from the study without any penalty. The ethical principle of fairness, privacy, confidentiality, and anonymity as well as the participants' rights to voluntarily participate in the study were adhered to. Collected data were stored on the laptops with a password to deny access to the people who were not a part of the project.

### 2.5. Data Collection

Data were collected through unstructured interviews to gain a detailed narrative of participants on their experiences of effective communication between heath care professionals and relatives in public hospitals. This was conducted after the patient was discharged as during admission, it was scary to talk about the patient's condition since the outcome was uncertain. Therefore, interviews were conducted by experienced qualitative researchers on the telephone at home during the participant's free and convenient time. One central question was asked "Can you please share your experiences regarding communication with health care professionals during the era of COVID-19 at public hospitals?" Probing was carried out by focusing on barriers to effective communication between health care professionals and patients in public hospitals. The interviews were conducted by researchers in the participants' local language (Tshivenda and Xitsonga), and each lasted between 30–45 min. Permission to use the voice recorder was obtained from the participants.

### 2.6. Data Analysis

The narrative data from the in-depth individual interviews were analyzed qualitatively using Tesch's open coding method, as postulated by [10]. Collected data were interpreted and described in the early phases of data analysis. The recorded interviews were translated

into English by the language expert, and transcribed word by word so that the information could be accessed by everyone who did not understand the local language used. Therefore, English was found to be an international communication language used globally. Among the researchers, there were experts who fluently understood the local language used (Tshivenda and Xitsonga) and were able to provide an authentic translation. All transcripts were read to provide meaning, and a list of similar topics or ideas were put together. Data were grouped per themes and sub-themes, and field notes were also coded and categorized. Ideas that were related to the main theme were grouped as sub-themes to enhance the flow of ideas. A literature control was conducted to control the results of the study.

*2.7. Trustworthiness*

The criteria for ensuring trustworthiness as outlined in [6], Polit and Beck, were adhered to. Credibility was ensured through prolonged engagement during data collection. Furthermore, the voice recorder was also used to ensure credibility as the participants were made aware that they were voice recorded and gave consent to the recording. The appointment was made and the purpose of the study was highlighted by telephone. During the interviews, the researcher spent time with the participants listening and recording them as they were interviewed. Each individual person was interviewed until they related all of their stories. However, all of the participants were interviewed to the point at which there was data saturation. A member check was also conducted by the research team to validate the truth and to confirm the findings. This was carried out by taking back the findings of the study to the participants to add information and to correct obvious errors, if any. The recorded interviews were transcribed verbatim.

**3. Results**

The study had 11 participants of which four (4) were relatives and seven (7) were patients, as indicated in Table 1. Data were collected during lockdown level 3 from patients who were admitted to the hospital during May to July 2020. Data were collected after the patients were discharged. Participants were interviewed using a telephone in their local languages, which were Tshivenda and Xitsonga.

**Table 1.** The participants' demographic data.

| Participant Number | Age | Gender | Patient or Relative | Language |
|---|---|---|---|---|
| 1 | 26 | Male | Patient | Tshivenda |
| 2 | 30 | Male | Patient | Tshivenda |
| 3 | 39 | Female | Patient | Xitsonga |
| 4 | 28 | Female | Relative | Tshivenda |
| 5 | 20 | Male | Relative | Tshivenda |
| 6 | 40 | Female | Patient | Xitsonga |
| 7 | 35 | Female | Patient | Xitsonga |
| 8 | 28 | Male | Patient | Xitsonga |
| 9 | 44 | Male | Patient | Tshivenda |
| 10 | 51 | Female | Relative | Xitsonga |
| 11 | 47 | Female | Relative | Tshivenda |

The research results outlined three themes and seven sub-themes, as displayed in Table 2, which were analyzed using Tesch's open coding. Themes were created from the data and categorized as follows: poor communication of diagnosis and prognosis, inadequate information of the disease process, and inconsistencies in the dissemination of COVID-19 protocols.

**Table 2.** The themes and sub-themes derived from the collected data.

| | | |
|---|---|---|
| 1. | Poor communication of diagnosis and prognosis | 1.1 Failure to come up with specific diagnosis |
| 2. | Inadequate information of the disease process | 2.1 Lack of knowledge<br>2.2 Avoidance behavior by health care professionals |
| 3. | Inconsistencies in dissemination of COVID-19 protocols. | 3.1 Negative view or experience on protocol use<br>3.2 Compromised visitation hours<br>3.3 Discontent with hospital communication protocol<br>3.4 Positive experience on protocol use |

*3.1. Theme 1: Poor Communication of Diagnosis and Prognosis*

Participants from this study revealed that the communication of diagnosis was not properly carried out. Investigations were conducted but never communicated to the participants so they were left in the dark. The new pandemic brought confusion to both health care professionals and the patients since little was known about the disease.

Sub-Theme 1.1 Failure to Come Up with Specific Diagnosis

One participant indicated that he was sent to a ward without a proper diagnosis, thinking that perhaps the real diagnosis would be communicated after the results of the tests ordered.

This was confirmed by Participant 6, who said "I went there only coughing, suddenly I was sent to the other ward, I was there for close to a month admitted there, some tests were done, and I was not told of my results. Secondly, they were not explaining clearly why I was there, I just heard is COVID-19 ward but I did not have that disease".

Nurses are supposed to act as advocates for patients, however, the participant indicated that the nurses also left them in the dark, and that nurses are supposed to make nursing diagnoses and reassure patients.

Participant 3 added by saying "Nurses were not telling me anything. I asked them about the results and they told me to wait for the doctor. This is really depressing me. Eish . . . . I was not comfortable as I didn't know exactly what the problem was".

Health care professionals tended to underestimate the health worker–patient communication, so the patients were dissatisfied when open communication was not coming through or taking place.

*3.2. Theme 2: Inadequate Information of the Disease Process*

The participants needed to be given full and accurate information about the disease process, however, the participants were given little or no information at all about the COVID-19 disease processes. Patients need to be aware of their right to information; they should not be passive and need to be involved in the decision-making regarding their health.

3.2.1. Sub-Theme 2.1: Lack of Knowledge

When participants receive information about the disease process, it allows them to know what to do and what to avoid. Furthermore, it helps the participants cope well as they know what to expect.

This is what Participant 1 said: "I was not given information about this. Only when I asked, a person will just answer in one sentence and leaves. COVID-19 is scary, I don't know but it's like we were being ignored".

It was also noted that health workers usually tried to avoid patients or discourage patients from voicing their concerns and expectations as well as requests for more information, as if patients should just be passive and vulnerable and not voice their expectations.

Participant 9 said "Yes, I know the symptoms, but I cannot say that I know everything about COVID-19. It needs someone who is knowledgeable and patient enough to can explain information that will be of help to me".

A lack of sufficient explanation results in poor patient understanding, and a lack of consensus between the doctor and patient may lead to therapeutic failure.

### 3.2.2. Sub-Theme 2.2: Avoidance Behavior by Health Care Professionals/Reluctance to Disclose Patient Condition by a Health Care Professional

This study revealed that some health care professionals were unwilling to disclose the health condition as they also did not have sufficient information about the disease and the treatment protocols as trials were still ongoing.

This was attested to by Participant 2, who expressed himself as follows: "You know, those nurses were not serious (shaking the head). I have been asking about my health condition and they were telling me to wait for the doctor. Some just ignore the question if you happen to ask them something . . . This was not right really. Who must tell me then because I relied on them?"

Breaking bad news to patients is a complex and challenging communication task in the practice of medicine, but especially so during the COVID-19 pandemic, where the disease was new, and research was still underway to understand the pathology of the virus.

One relative, Participant 4, who appeared very angry, said "I don't know why my child was admitted because I did not get any information about her condition, and no one was willing to tell me something. Health care professionals were excusing themselves by being busy all the time. I think I was ignored, and it is really disturbing my mind".

### 3.3. Theme 3: Inconsistencies in Dissemination of COVID-19 Protocols

The Department of Health (DoH) in South Africa developed COVID-19 protocols that were meant to reduce the spread of infections in health care settings and in the communities at large. However, the use of the protocols was different from one institution to the other. Others would allow the relatives to see their patients through the windows, others did not allow them at all, whereas some institutions would allow relatives to call the patients via a hospital telephone.

### 3.3.1. Sub-Theme 3.1: Negative View or Experience on Protocol Use

Participants in this study indicated that the protocols were not communicated with them during admission. This is how they expressed their negative experience of the COVID-protocols:

Participant 10 (relative) said, "Hey this policy makes it so difficult for us as a family as we cannot rely on nurse's report without seeing your loved one. Especially when you listen to social media. Each time you hear your phone ring you just expect bad news".

The hospitals failed the patients and relatives because they could not allow their relatives to visit, so the relatives were always anxious and anticipating bad news because of the reports of mortalities in the media.

### 3.3.2. Sub-Theme 3.2: Compromised Visitation Hours

Visitation time were also affected, as some institutions allowed it a maximum of two people per visit, while other institutions denied access to all visitation. This was supported by Participant 8, who said:

> "My husband called me from the hospital gate and said he was coming to see me, but he was denied access to come to the ward where I was. It really surprises me as to why it is like that. He was told that all patients are not to be visited. I wanted to go and meet him, but they also refused when I wanted to go out, and they said that I will infect other people. I think the hospital was supposed to have a way to see our relatives. It is not good to be in a serious condition and I do not

see my family just to verbalize my feeling because there is no one to verbalize my feelings to".

It is necessary for hospitals to have a workstation that is staffed with health care professionals who understand ethical principles to be able to communicate with relatives and communities.

### 3.3.3. Sub-Theme 3.3: Discontent with Hospital Communication Protocol

The hospitals failed to practice Batho–Pele principles, even though the department is vocal about it. During the pandemic, the institutions should have opened telephone lines to keep the community informed such as in times of disasters and assist patients to talk to their relatives for assurance. Participant 7 had this to say:

> "I wanted to contact my children at home. Unfortunately, I did not have my cell phone with me. I asked nurses to call for me using hospital telephone. I was so shocked when they refused to call for me, saying that the phone is for work related things only. Yoh, it is really frustrating, staying at the hospital without seeing or even talking to them".

In health care settings, communication represents a fundamental clinical skill that involves the establishment of a therapeutic relationship, understanding the patient's perspective, and exploring their thoughts and emotions, because during pandemics, patients experience stress and fear from the pandemic.

### 3.3.4. Sub-Theme 3.4: Positive Experience on Protocol Use

Some participants viewed the use of COVID-19 protocols during patient's hospitalization as beneficial to them because they were being protected from infection. Even though it was difficult for them to not see their relatives, they were happy that it was for their own good to protect and preserve their lives. This was confirmed by some relatives who said:

Participant 11 (relative) said: "For the sake of protection from getting infection is good, but this was too much. Is too painful. I think they need to device the other means which is safe for both of us (patient and relatives) for us to see our family member who is in dire need of support from us. You know, I am not against the policy, but I have experience something I never experience in my life".

Participant 5 (relative) added by saying that: "There was nothing wrong with the protocol policy, what was wrong is when nothing was communicated well in advance. Yes, we are to be protected from this thing that is killing us but now, what if our patient dies, are we to be surprised by his death, then we will be called?"

Accurate knowledge of COVID-19 from reliable sources is important and practicing and adhering to government advice helps in protecting vulnerable communities.

## 4. Discussion

Poor communication of diagnosis, as one of the themes highlighted above, was seen as a major barrier for the admitted patients and their relatives. Since the findings of the study indicated that health care professionals failed to provide an accurate diagnosis or communicate the prognosis to the participants, this aggravated uncertainty and unnecessary anxiety to the patients and relatives. The study conducted by [11] indicated that an appropriate expression of uncertainty can lead to strong health care professional and patient and family relationships. The participants had to wait for test results and were still not provided with any information or communication, which further strained their communication. Furthermore, the patients felt that they lacked knowledge, as nothing about the disease and disease process was communicated to them. On the other hand, the nurses were seen as not having enough information, as disclosed by the patients and relatives, and as a result, they tended to ignore the questions raised by both the patients and relatives. This was supported by the study conducted by [12], who revealed that nurses caring for COVID-19 patients were also psychologically affected. Contrary to the

findings of this study, this might be the other reason for the health care professionals not fully paying attention to the patients despite the issue of a lack of knowledge. Kapil, Verma, and Sareen (2019) [13] asserted that poor quality doctor–patient communication was found to be associated with patient dissatisfaction, reduced treatment adherence, and poor health outcomes. Furthermore, [14] indicated that poor or limited communication may be related to inadequate formal training in communication skills, and that this accountability has not been formally integrated into the curriculum or continuing education of many medical schools [15]. Al-Zahrani et al. (2015) reported that very few general practitioners always involve the patient in decision-making or discuss goals of consultation with their patients. Patients are human beings and need to be involved in their care; failure to be involved results in ineffective care. Patients have consistently shown that they want better communication with their doctors. Information exchange is the dominant communication model, and the health consumer movement has led to the current model of shared decision-making and patient-centered communication [16].

Shiraly, Mahdaviazad, and Pakdi (2021) [17] indicated that patients had different needs—physical, psychological, social, and spiritual ones—that should be identified and addressed by health care professionals through actively listening to patients and the appropriate gathering of relevant information. Open, effective, and productive communication between patients and their health care team is key to producing good patient health outcomes. A doctor's communication and interpersonal skills encompass the ability to gather information to facilitate accurate diagnosis, counsel appropriately, give therapeutic instructions, and establish caring relationships with patients [18]. The doctor–patient interaction should be investigated as a complex process because this is the most important factor to develop understanding if quality care is needed. This is achieved through proper communication. Once the aspect of communication is overlooked, serious potential pitfalls occur, especially in terms of the patients' understanding of their prognosis, purpose of care, expectations, involvement in treatment, etc. [19]

A lack of accurate information about the COVID-19 virus was another cause for poor patient–health care professional communication as the health care team were struggling on daily basis with the new information about the genotype of the virus, variants, mutations, pathophysiology, and clinical treatment trials. When clinical researchers thought they had established the treatment guidelines, the virus changed its trajectory, resulting in physicians not having enough information about the disease process and how to communicate accurate facts to patients who were anxious and distressed [20]. Mansour, Rallapalli, Baidwan, Razai, and Abou-Abbas (2022) reported in their study that 349 (87.5%) participants had good knowledge about the COVID-19 disease and COVID-19 vaccines, however, poor knowledge was observed for questions concerning the nature of disease (52%) and treatment of the disease (59.9%). The participants further indicated that health care workers were using the avoidance strategy to mask communication [21]. Lancet (2020) reports that some physicians preferred not to be entirely honest with patients regarding the possible severity of the disease's symptoms, as this could trigger panic in the patients regarding their health conditions. Furthermore, [17] indicates that some physicians expressed reservations regarding the value of sharing pertinent information with patients because they believed that pandemic diseases were not fully understood, meaning that information had to be shared with the public carefully. This study revealed that the participants were not knowledgeable about the COVID-19 pandemic, which is in contrast to the study conducted by [22], who indicated that their participants displayed knowledge regarding COVID-19 and the preventative measures thereof.

Participants in the study reiterated the inconsistencies in the COVID-19 protocols in place at the different public hospital institutions. In some hospitals, the health workers displayed a humanitarian attitude while at other institutions, they did not show any empathy toward the patients and relatives. According to African culture, individuals need and require support from one another, so the culture is more of collectivism than individualism. The patients and relatives wanted to visit and communicate with their loved

ones, but structural and administrative barriers prevented that. The participants indicated that they were not allowed to visit, while others could see their loved ones through the windows. These are some of the limitations that occurred due to different interpretations of the protocols. This is supported by [21], who reported that more than half of the participants (56%) indicated that the policies implemented to handle COVID-19 by their public health agencies were insufficient or disorganized. Furthermore, the authors suggested the need for local decision-making at the local level according to the degree of incidence. With regard to the inter-departmental communication, the physicians recommended standardizing and centralizing the distribution of pandemic-related communications. This is important that the same protocols should be regional and context-based rather than each institution managing things differently. Implementing a uniform approach in the region is crucial as the efforts of one public institution, or many institutions working in silos, cannot eliminate the threat and the psychological impact of a pandemic [23].

## 5. Limitations of the Study

The study was conducted during a global new pandemic that was, and still is, life threatening. Therefore, there were some factors that limited the researchers from operating as planned. This included the method used to interview the participants, as some non-verbal cues could not be observed. Lack of sufficient information such as the protocol used was not clearly disseminated to the patients and relatives.

## 6. Implications for Practice

Patients who recuperate from COVID-19 should not be discharged straight home without counselling for mental health support due to the psychosocial consequences of COVID-19. Health care services should develop a support system for patients and relatives affected by pandemics where it is difficult to predict the health outcomes of loved ones and when the situation will return to normal.

## 7. Conclusions

Good communication between patients, family caregivers, and the health care team is very important in public health care institutions as patients and families require a lot of information and choose to make decisions regarding care. The participants described their experience with COVID-19 in negative emotive language and agreed that the pandemic had led to increased mental stress due to poor communication protocols and skills by the health care workers. Pandemics usually cause conflicts. Therefore, part of the preparedness for future health pandemic crises should include proper communication skills and the standardization of how to communicate protocols.

**Author Contributions:** D.U.R. and N.S.R. contributed to the study design and writing the draft. T.R.L., M.M., N.D.N., T.M., A.R.T. and M.S.M. analyzed and interpreted the collected data. H.N.S., T.E.M. and M.S.M. critically appraised the draft of the manuscript whereas J.L.M., S.A.M., M.L.N. and K.G.N. edited the final draft. All authors have read and agreed to the published version of the manuscript.

**Funding:** The project was financed by the University Research Fund.

**Institutional Review Board Statement:** Ethical considerations were insured by obtaining ethical clearance (SHS/20/PDC/19/0608) from the University of Venda Ethics Committee as well as permission to conduct the study from Board members of the selected hospitals and the participants.

**Informed Consent Statement:** Informed consent was obtained from all subjects involved in the study.

**Data Availability Statement:** The data are available on request.

**Acknowledgments:** The authors really appreciate the cooperation offered by the Chief Executive Officers and nursing managers who assisted with essential logistic support and the participants who consented to being part of the study.

**Conflicts of Interest:** The authors declare no conflict of interest.

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
