# Peer review of "Barriers to Effective Communication between Patients, Relatives, and Health Care Professionals in the Era of COVID-19 Pandemic at Public Hospitals in Limpopo Province"

_2673-527X, doi:10.3390/jor3010004_

Round 1

Reviewer 1 Report

Comments in deatils are added to the text (attached). I suggest to make corrections in grammar and stylystics (minor), also to erase gaps between words in some sections.

Abstract- final statement is incorrect, not supported by the aim and the research itself. Therefore I suggest to rethink and rewrite it.

Introduction- suggesting to enhance. 

Citation in the text- suggesting to place the citation numbers at the right spot in the whole text.

 Also I would suggest to ammend the references as this topic is wildelly researched.

The manuscript could be enhanced by implications for practice, suggestions, etc.

Author Response

Thanks for the comments suggested. They were helpful

Reviewer 2 Report

Congratulations for Authors

Author Response

Thank for your recommendations

Reviewer 3 Report

Dear Authors, having enjoyed reading the manuscript, please find my comments and feedback below for your consideration.

Barriers to Effective Communication between Patients, Relatives, and Health Care Professionals in the Era of COVID-19  Pandemic at Public Hospitals in Limpopo Province

A qualitative descriptive study.

Abstract

Reads clearly, does use emotive language such as ‘despair’ which is not supported by the text. In general emotive words should only be used when a direct quote from a participant – researchers ideally use research language.

The conclusion could be clearer and stronger if it stated the exact benefits of full communication rather than stating it is ‘imperative’ to give full communication.

Key words

Barrier and era are not helpful key words, lacking context; consider some key words related to the methods as well as the topic.

Introduction

Please add the words ‘…with patients’ to the end of the first sentence of the first paragraph, that’s the main population of interest to this study. Please add references to support statements, eg 36-37, 43-46, 46-48, 48-51 – these are all from a source which needs to be acknowledged.

Line 79-83 that’s a good explanation and justification. 

Line 85

‘A qualitative approach using explorative, descriptive, and contextual design were used.’ That’s a very long description, as I read, I need to see how these terms are operationalised in the methods with appropriate depth of description so a reader can duplicate each aspect of the methods.

NOTE: having read the methods, it seems these terms are not fully utilised, please keep it clear and simple by using the term ‘qualitative descriptive’ – the rest of the terms such as ‘explorative’ and ‘contextual’ are not helpful or accurate.

Line 87 – this repeats line 85, please edit to avoid overt repetition

Line 89-92 this is unclear and a bit confusing, please rephrase to give a clear justification of the choice of methods. 

Line 94 – the study was conducted ‘in’ four selected public hospitals…

Population

Not sure why there is a need to define a population, equally unclear why ‘all patients who were admitted ...’ are relevant, since this study used purposive sampling. What we are interested in was how the sample was identified.  

Sampling

Purposive sampling is by definition ‘nonprobability sampling’; no need to use both terms, and purposive is more descriptive, so please use that term.

This section should also describe the preferred sample characteristics and HOW those people were identified and invited to join the study – at the moment the sampling section is a general description of sampling, without specific information to help a reader understand this study – please make it specific!  We need to know ‘how’

Ethics

The statement on line 115-6 is good, but needs expanding – where and how is interview data stored post interview? Were patients coded during or after interviews? Why was consent verbal rather than signed? Was an information sheet given to patients about the study? What steps aside from a pseudonym were taken to protect confidentiality and anonymity?

Data collection

This could be clearer – unstructured interviews is great, but was it collected during hospital stay as well as following discharge? Where and how were the interviews conducted, was the interviewer an experienced qualitative researcher? What setting, and how was patient confidentiality and anonymity protected? 

This section is vital to the credibility of the study, be detailed and specific please. 

Data analysis

Why were interviews translated to English? Was there back translation to verify accuracy? Who did the translation and what skills/experience qualified them to be the translator?

Its not clear what ‘topics’ are or how they were determined to be authentic.

What is a cluster of topics? Please clarify, we need more detail on the methods. 

How was ‘similarity’ determined – ie what decision making process by how many researchers

If topics were clustered, then data grouped, are these terms the same or different steps in the analysis? 

How were categories created?

Please describe the process for creating themes and sub-themes – we need to know about both of these processes.

Trustworthiness

The methods do not describe any pre-meetings with the participants.

Individual interviews are not conducted to data saturation, they are conducted until the person has ‘told their story’ – data saturation is about the total number of interviews and can only be claimed where there is concurrent data collection and data analysis. Since this study collected data before analysis, there is no clear way to claim data saturation, and this is a specific limitation that needs to be reported in the discussion. 

Who undertook the member check and how did they do it? Please describe

Why does a voice recorder ensure credibility? What other steps were taken for credibility.

Transferability is not established through thick description, and as this is a descriptive study, thick description is not relevant, this sentence should be removed.

Results

The results states patients were interviewed telephonically which means they were called on a phone. This should be stated in the methods! Its an important point readers need to know before they get to the results.

Equally, why not say ‘phone’ or ‘telephone’ -its clearer and simpler to use common language and this makes the work more accessible to more people. Adding technical words when a regular word will do does not help readers.

The methods talk about clusters and categories, but there is no mention of these in the results, nor is it clear how clusters and categories contributed to each theme. 

Please note – themes do not ‘emerge’ – themes are created by authors through analytic processes. Please avoid any use of the term ‘emerge’

The themes appear to be supported by participant quotes which is good reporting. 

Discussion

The key findings are the themes, please start by referring to the major themes before moving into clinical application. 

The rest of the background reads like a brief literature review but does not appear to fit well with the results – this could be addressed by more clearly aligning the discussion with the results on a theme by theme basis, and drawing parallels between existing literature and the findings of this study (using sub-headings)

There needs to be a sub-heading for ‘Study Limitations’ where the limitations specific to the methods are recorded. I have noted several limitations in the above notes, and those as a minimum should be added to the limitations section.

Conclusions

The conclusions are too vague, they make a suggestion which does not concisely fit the results of the study. This research did not investigate ‘a best strategy’ – focus the last sentence of the conclusions on what the participants wanted as a mechanism for activating the major themes please.

Author Response

We, as authors really thankful for the valuable comments given. They really improved our manuscript

Author Response

Thanks for the comments given. They were so helpful 

Round 2

Reviewer 3 Report

I have no further feedback, the paper has improved readability for the core concepts.

Reviewer 4 Report

i appreciated your revisions. They lended clarity to your manuscript. I suggest that you do a final proof reading of your manuscript but otherwise I am satisfied with your changes.